# Theory of optical responses in clean multi-band superconductors

Junyeong Ahn [1,2 ✉] & Naoto Nagaosa[2,3 ✉]

Electromagnetic responses in superconductors provide valuable information on the pairing symmetry as well as physical quantities such as the superfluid density. However, at the superconducting gap energy scale, optical excitations of the Bogoliugov quasiparticles are forbidden in conventional Bardeen-Cooper-Schrieffer superconductors when momentum is conserved. Accordingly, far-infrared optical responses have been understood in the framework of a dirty-limit theory by Mattis and Bardeen for over 60 years. Here we show, by investigating the selection rules imposed by particle-hole symmetry and unitary symmetries, that intrinsic momentum-conserving optical excitations can occur in clean multi-band superconductors when one of the following three conditions is satisfied: (i) inversion symmetry breaking, (ii) symmetry protection of the Bogoliubov Fermi surfaces, or (iii) simply finite spin-orbit coupling with unbroken time reversal and inversion symmetries. This result indicates that clean-limit optical responses are common beyond the straightforward case of broken inversion symmetry. We apply our theory to optical responses in FeSe, a clean multi-band superconductor with inversion symmetry and significant spin-orbit coupling. This result paves the way for studying clean-limit superconductors through optical measurements.

[1] Department of Physics, Harvard University, Cambridge, MA, USA. [2] RIKEN Center for Emergent Matter Science (CEMS), Wako, Saitama, Japan.
[3] Department of Applied Physics, The University of Tokyo, Bunkyo, Tokyo, Japan. ✉email: junyeongahn@fas.harvard.edu; nagaosa@riken.jp

Optical studies have been very important in super-conductivity research since the superconducting gap was first observed by far-infrared optical measurements[1]. Not only does the optical absorption gap directly reveal the super-conducting gap size, but also the loss of spectral weight of the optical conductivity in the superconducting transition shows the superfluid density[2–4]. Optical responses in superconductors are well understood by the dirty-limit theory of Mattis and Bardeen[5] and its extensions to arbitrary purity[6,7]. Impurity is essential in the Mattis–Bardeen theory because Bogoliubov quasiparticles cannot be excited by uniform light when momentum is conserved in the Bardeen–Cooper–Schrieffer (BCS) model[8,9]. In this paradigm, optical responses are due to impurity scattering and correspond to the Drude responses remaining in the super-conducting state. They are thus completely described within a single-band model such as the BCS model and approaches the Drude formula as the photon energy increases above the gap [Fig. 1].

On the other hand, there have been cumulative studies revealing the relevance of multiband effects in superconductivity. Strong gap anisotropy and multiple gap signatures due to orbital-dependent pairing have been observed in various super-conductors, including elemental metals Nb, Ta, V, and Pb[10,11], compound $MgB_2$[12], strontium titanates[13], iron pnictides and chalcogenides[14–16], and heavy fermion compounds[17,18]. Multi-band effects are also considered to be important in the super-conductivity of strontium ruthenates[19,20], some half-Heusler compounds with $J = 3/2$ degrees of freedom[21–23], and twisted bilayer graphene[24–26]. This raises the question of whether multi-band effects can modify the optical responses. However, there has been no observation of a significant deviation from the Mattis–Bardeen theory in any materials.

In this work, we challenge the Mattis–Bardeen paradigm by showing that a significant portion of the far-infrared optical response in a clean multiband superconductor FeSe is due to intrinsic momentum-conserving excitations. We show that such intrinsic optical excitations are allowed by multiband effects. To establish the criteria for nonzero intrinsic responses systematically, we present a tenfold way classification of optical excitations as well as selection rules due to unitary symmetries. Here, the tenfold way classification is by three symmetry operations $\mathfrak{T}$, $\mathfrak{C}$, and $S$ that leave momentum invariant[27], whereas the original tenfold classification by Altland and Zirnbauer[28,29] is by three spatially local symmetries, including time reversal $T$, particle–hole conjugation $C$, and chiral $S$ symmetries [see Table 1]. Since $T$ and $C$ reverses the momentum, the combination of them with spatial inversion $P$ (or any other momentum-reversing unitary operation) defines $\mathfrak{T}$ and $\mathfrak{C}$. In this classification, $\mathfrak{C}$ symmetry is the key player that imposes a new selection rule. We find that the absence of intrinsic optical excitations in single-band models can be attributed to $\mathfrak{C}$ symmetry in the superconducting state.

As real materials are always accompanied by disorder, intrinsic responses coexist with disorder-mediated responses. A super-conductor is considered to be clean when the mean free path $l$ is larger than the superconducting coherence length $\xi_0$ and to be dirty when $l$ is smaller than $\xi_0$. We show that the crossover from disorder-mediated to intrinsic optical responses occurs at a very clean regime $l \sim (k_F \xi_0)^2 \alpha^{-2} \xi_0 \gg \xi_0$, as illustrated in Fig. 1b, where $k_F$ is the Fermi wave number, and $0 \leq \alpha \leq 1$ is a degree of multi-band pairing explained below. When $l$ goes above this value, the optical conductivity follows the $\omega^{-1}$ behavior of the intrinsic response, deviating from the Drude-like $\omega^{-2}$ behavior [Fig. 1d, e]. We discuss the optical response of superconducting FeSe, which is closest to this crossover regime.

## Results

**Setting.** Our theory is based on the mean-field theory of super-conductors. We assume uniform illumination of light at zero temperature and the conservation of momentum. In momentum

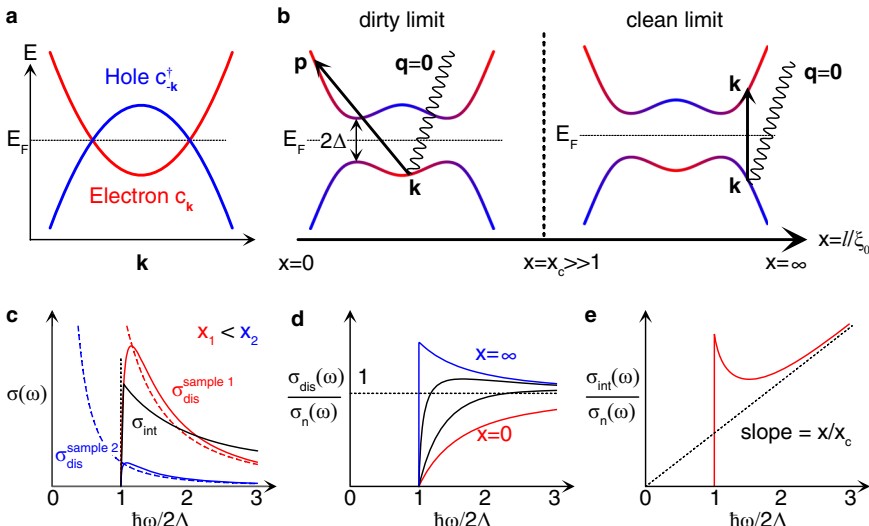

**Fig. 1 Dirty and clean optical responses. a** Band structure in the Bogoliubov de-Gennes (BdG) formalism. Superconducting pairing opens the bandgap at the Fermi level $E_F$ by mixing electron (red) and hole (blue) bands as shown in (**b**). $c_k$ and $c^\dagger_{-k}$ indicate the electron and hole annihilation operators, respectively. **b** Optical excitations in dirty and clean limits by spatially uniform light (**q = 0**) across the superconducting gap $2\Delta$. The momentum transfer **p** − **k** in the left figure is supplied from the impurity potential. The crossover from dirty to clean optical responses occurs when the mean free path $l$ exceeds the superconducting coherence length $\xi_0$ by a factor $x_c = (k_F \xi_0)^2 \alpha^{-2}$. Here, $k_F$ is the Fermi wave number, and $\alpha$ is a degree of multiband pairing (see the text above Eq. (6) for its more precise definition). The red and blue colors schematically represent the electron–hole band mixing in the superconducting state. **c** Real part of the optical conductivity in clean systems ($x > 1$). $\sigma_s = \sigma_{dis} + \sigma_{int}$ in the superconducting state is the sum of disorder-mediated and intrinsic parts. $\sigma_{int} \propto \omega^{-1}$ is insensitive to $x$ while $\sigma_{dis} \propto x^{-1}\omega^{-2}$ depends significantly on $x$. The Drude conductivity in the normal state $\sigma_n$ is also shown as dashed lines. **d**, **e** Ratio of superconducting and normal state conductivities. **d** Disorder-mediated part. **e** Intrinsic part.

**Table 1 Tenfold way classification of the lowest optical excitations in superconductors.**

| EAZ class | $\mathfrak{T}^2$ | $\mathfrak{C}^2$ | $S^2$ | Lowest excitation | BFS stability |
|---|---|---|---|---|---|
| A | 0 | 0 | 0 | Yes* | $\mathbb{Z}$ |
| AI | 1 | 0 | 0 | Yes* | $\mathbb{Z}$ |
| AII | −1 | 0 | 0 | Yes* | $\mathbb{Z}$ |
| AIII | 0 | 0 | 1 | Yes | 0 |
| D | 0 | 1 | 0 | Yes | $\mathbb{Z}_2$ |
| BDI | 1 | 1 | 1 | Yes | $\mathbb{Z}_2$ |
| C | 0 | −1 | 0 | No | 0 |
| CI | 1 | −1 | 1 | No | 0 |
| DIII | −1 | 1 | 1 | Yes | 0 |
| CII | −1 | −1 | 1 | Yes | 0 |

Anti-unitary $\mathfrak{T}$, anti-unitary anti-symmetry $\mathfrak{C}$, and unitary anti-symmetry $S$ operators that do not change the momentum define ten effective Altland–Zirnbauer (EAZ) symmetry classes at a given generic momentum. 0 in the second set of columns indicates that no corresponding symmetry exists within the eigenspace of interest. When both $\mathfrak{T}$ and $\mathfrak{C}$ symmetries exist, $S = \mathfrak{T}\mathfrak{C}$ in classes BDI and CII ($S = i\mathfrak{T}\mathfrak{C}$ in classes CI and DIII) when we choose the convention $\mathfrak{T}\mathfrak{C} = \mathfrak{C}\mathfrak{T}$. In classes A, AI, and AII, the lowest possible excitation energy within an eigenspace may not correspond to the direct superconducting gap, because the states with the lowest positive and the highest negative energies may have different symmetry eigenvalues. The asterisk (*) in the third column means that the excitation cannot occur when there is only one band in an eigenspace. The last column shows the stability of Bogoliubov Fermi surfaces (BFSs), meaning nodal surface/line/point in 3D/2D/1D superconductors.

space, the single-particle mean-field Hamiltonian has the Bogoliubov-de Gennes (BdG) form

$$H(\mathbf{k}) = \begin{pmatrix} h(\mathbf{k}) & \Delta(\mathbf{k}) \\ -\Delta^*(-\mathbf{k}) & -h^T(-\mathbf{k}) \end{pmatrix} \quad (1)$$

in the basis of the Nambu spinor defined by $\hat{\Psi} = (\hat{c}_{\rho s \mathbf{k}}, \hat{c}_{\rho s -\mathbf{k}}^\dagger)^T$, where $\hat{c}_{\rho s \mathbf{k}}$ is the electronic quasiparticle annihilation operator with orbital $\rho$ and spin $s = \uparrow, \downarrow$ indices. Here, $h(\mathbf{k})$ is the normal-state Hamiltonian, and the pairing function $\Delta(\mathbf{k}) \propto \langle c_\mathbf{k} c_{-\mathbf{k}} \rangle$ satisfies $\Delta(\mathbf{k}) = -\Delta^T(-\mathbf{k})$ due to Fermi statistics of electrons. The BdG Hamiltonian always has particle–hole symmetry $CH(\mathbf{k})C^{-1} = -H(-\mathbf{k})$ under $C = \tau_x K$, where $\tau_x$ is a Pauli matrix for the particle–hole indices and $K$ is the complex conjugation operator.

The electromagnetic field couples to the normal-state Hamiltonian through the minimal coupling $\mathbf{k} \rightarrow \mathbf{k} + \frac{q}{\hbar} \mathbf{A}$, where $q = -e$ (+e) for the electron (hole) sector. It follows that the velocity operator is

$$V^a(\mathbf{k}) = \frac{1}{e} \frac{\partial H}{\partial A_a}\bigg|_{\mathbf{A}=0} = \frac{1}{\hbar} \begin{pmatrix} \partial_{k_a} h(\mathbf{k}) & 0 \\ 0 & \partial_{k_a}[h^T(-\mathbf{k})] \end{pmatrix}. \quad (2)$$

Matrix elements of this operator are important in our analysis because they describe the transition amplitudes. In the clean limit, the real part of the optical conductivity tensor is given by

$$\sigma^{ca}(\omega) = \frac{\pi e^2}{2\hbar\omega} \int_\mathbf{k} \sum_{n,m} f_{nm}(\mathbf{k}) V_{nm}^c(\mathbf{k}) V_{mn}^a(\mathbf{k}) \delta(\omega - \omega_{mn}(\mathbf{k})), \quad (3)$$

where $\omega$ is the frequency of light, $f_{nm} = f_n - f_m$ is the difference between the Fermi distribution of the $n$th band $f_n$, $V_{mn}^a = \langle m | V^a | n \rangle$, and $\omega_{mn} = \omega_m - \omega_n$, where $H|n\rangle = \hbar\omega_n|n\rangle$[9,30]. The delta function is replaced by the Lorentzian distribution when the mean free path is finite.

**Selection rules.** Equation (3) is positive-semidefinite when $c = a$. Therefore, interband transitions are completely forbidden only when symmetries impose $V_{mn}^a(\mathbf{k}) = 0$ at every $\mathbf{k}$[30]. The relevant symmetry operators should be $\mathbf{k}$-local ($\mathbf{k} \rightarrow \mathbf{k}$). A unitary symmetry imposes selection rules by $\lambda_m(\mathbf{k}) = \lambda_V(\mathbf{k})\lambda_n(\mathbf{k})$, where $\lambda_{m,n}$ and $\lambda_V$ are symmetry eigenvalues of $m,n$ states, and the velocity

operator, respectively. We always have $\lambda_V = 1$ because $\mathbf{k}$-local symmetry operations leave $V^a$ invariant, as one might expect because the velocity operator should transform like $\mathbf{k}$. The selection rules thus simply become

$$V_{mn}^a(\mathbf{k}) = 0 \quad \text{when } \lambda_m(\mathbf{k}) \neq \lambda_n(\mathbf{k}), \quad (4)$$

meaning that optical excitations are forbidden between two different eigenspaces [Fig. 2a].

Let us consider the transition between two states in the same eigenspace of the unitary symmetry group. The remaining $\mathbf{k}$-local symmetries come in three types: anti-unitary $\mathfrak{T}$, anti-unitary anti-symmetry $\mathfrak{C}$, and unitary anti-symmetry $S$, where anti-symmetry means that the operator anti-commutes with the Hamiltonian. They form ten EAZ symmetry classes[27–29,31] shown in Table 1. $\mathfrak{T}$ (or $\mathfrak{C}$) comes as a combination of $T$ (or $C$) with a $\mathbf{k}$-reversing unitary operator such as spatial inversion $P$ in any dimensions or twofold rotation $C_{2z}$ in two dimensions. $S$ is the combination $\mathfrak{T}\mathfrak{C}$ up to a phase factor. We find that only $\mathfrak{C}$-type symmetry can additionally exclude transition channels within an eigenspace of the unitary symmetry group. By using that $\mathfrak{C}$ is anti-unitary and that the velocity operator is invariant under $\mathfrak{C}$ as shown in the "Methods" section 1, we have

$$\langle \mathfrak{C} \cdot n\mathbf{k} | V^a(\mathbf{k}) | n\mathbf{k} \rangle = 0 \quad \text{when } \mathfrak{C}^2 = -1. \quad (5)$$

This constrains, in particular, the lowest-energy excitations, as illustrated in Fig. 2b, c. If bands are nondegenerate in each eigenspace, Eq. (5) indicates that the excitations across the gap are forbidden when $\mathfrak{C}^2 = -1$ [Fig. 2b]. See class C and CI in Table 1.

We find that the absence of optical excitations in single-band metal models, described by a two-band BdG Hamiltonian, can be attributed to the existence of $\mathfrak{C} = i\tau_y K$ symmetry. A single-band metal has the $\mathfrak{C}$ symmetry in the superconducting state, independent of the pairing symmetry, when it has symmetry $\xi(\mathbf{k}) = \xi(\mathbf{k})$ in the normal state, where $h(\mathbf{k}) = \xi(\mathbf{k})$ is the $1 \times 1$ Hamiltonian. Since the formation of Cooper pairs at the Fermi level requires such a symmetry relating $\mathbf{k}$ and $-\mathbf{k}$, it means that typically no optical excitations can occur in superconductors originating from single-band metals. One can extend this result to show the absence of optical excitations in multiband systems satisfying a generalized single-band pairing condition, the so-called zero superconducting fitness condition (see section 2 in the "Methods").

We have three ways of generating nontrivial optical excitations in an eigenspace. When bands are nondegenerate within an eigenspace, one can (i) break $\mathfrak{C}$ symmetry (EAZ class A, AI, AII, and AIII) or (ii) realize $\mathfrak{C}^2 = +1$ (class D and BDI). (iii) Or, when bands are Kramers degenerate due to $\mathfrak{T}$ symmetry satisfying $\mathfrak{T}^2 = -1$, lowest-energy excitations are generally allowed irrespective of the sign of $\mathfrak{C}^2$ (class DIII and CII). The first condition (i) just means breaking inversion symmetry when other unitary symmetries do not exist, which was demonstrated in ref. [30]. The second (ii) implies that the superconductor may host stable BFSs[27,32]. Since $\mathfrak{C}^2 = +1$ protects 0D $\mathbb{Z}_2$ topological charges, $\mathbb{Z}_2$-stable nodal surfaces/lines/points in 3D/2D/1D can appear after superconducting pairing on the Fermi surfaces, respectively, which we call as BFSs without distinguishing their dimension. Let us note that these are twofold degenerate BFSs. On the other hand, a stable nondegenerate BFS can appear in the EAZ classes A, AI, and AII. For instance, a superconductor with broken inversion and time-reversal symmetries can host stable BFSs[33–36]. Their stability is guaranteed by the change of the number of occupied BdG bands across the BFS, which is a $\mathbb{Z}$ topological charge. Since these classes correspond to the case (i), the symmetry protection of the stable BFSs, whether it is twofold degenerate or not, indicates that the lowest-energy optical excitations are possible. The last possibility (iii) is realized in $T$-

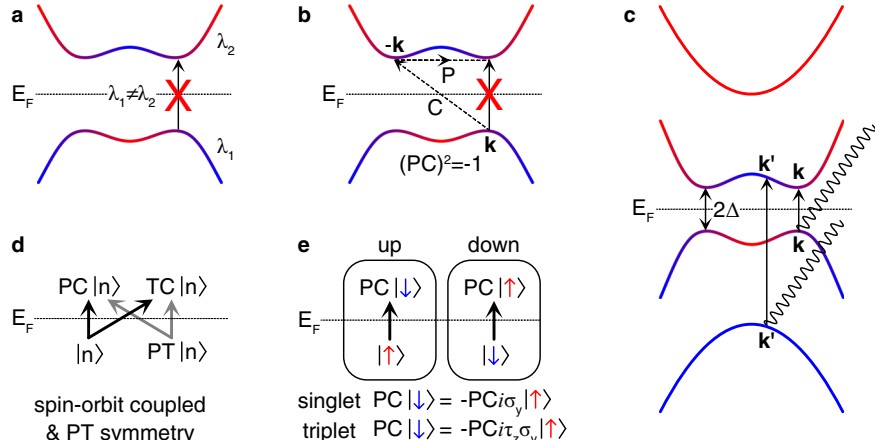

**Fig. 2 Selection rules in clean superconductors. a** Selection rule by unitary symmetry. $\lambda_{1,2}$ are eigenvalues of a **k**-local unitary symmetry operator. No optical excitation occurs between two states with different eigenvalues. **b** Selection rule by $\mathfrak{C}$ symmetry. The case with $\mathfrak{C} = PC$ is shown. No optical transition occurs between PC-related states when $(PC)^2 = -1$. **c** Optical excitation channels in $\mathfrak{C}$-symmetric superconductors in the clean limit. At low photon energies comparable to the superconducting gap $2\Delta$, the relevant excitations are spectrum-inversion-symmetric (SIS) ones, i.e., from energy $-E$ to $E$. For nondegenerate bands, they are transitions between $\mathfrak{C}$-related pairs. **d**, **e** Optical excitations in spin-degenerate systems with and without spin–orbit coupling, respectively. (here, the order of **d** and **e** has been changed in order to match the label in the figure.) In **d**, $\mathfrak{T} = PT$ symmetry with $(PT)^2 = -1$ imposes Kramers degeneracy. As a state $|n\rangle$ can be excited to one of two SIS states, $PC|n\rangle$ and $PT(PC|n\rangle) = TC|n\rangle$, the excitation from $|n\rangle$ is possible even when one transition channel, from $|n\rangle$ to $PC|n\rangle$, is blocked by $(PC)^2 = -1$. The same applies to the excitation from $PT|n\rangle$. In **e**, the boxes labeled up and down indicate the spin up and down eigenspace ($s_z = \hbar/2$ and $-\hbar/2$, respectively). Since $C$ reverses the spin (the anti-particle of a spin-up electron carries the down spin) while $P$ does not change the spin, $PC$ reverses the spin. Its combination with the spin rotation around the $y$ axis, which is $i\sigma_y$ for spin-singlet pairing, acts within a $s_z$ sector. For spin-triplet pairing, the spin rotation around the $y$-axis acts on the particle and hole sector with an opposite sign due to the spin carried by the Cooper pair, so the additional $\tau_z$ is introduced (see section 3 in the "Methods"). Optical excitations are forbidden when $\mathfrak{C}$ defined within a spin sector, which is $-iPC\sigma_y$ for singlet pairing ($-iPC\tau_z\sigma_y$ for triplet pairing), satisfies $\mathfrak{C}^2 = -1$. $E_F$ is the Fermi level in all figures.

and $P$-symmetric systems with spin–orbit coupling. Because of the twofold band degeneracy imposed by $\mathfrak{T} = PT$ symmetry, there are two excitation channels where $|n\rangle$ at energy $-|E|$ can be excited to $+|E|$ as shown in Fig. 2d. Even when one channel from $|n\rangle$ to $PC|n\rangle$ is excluded by $(PC)^2 = -1$, there exits another channel from $|n\rangle$ to $PT(PC|n\rangle) = TC|n\rangle$. In the absence of spin–orbit coupling, each spin sector forms nondegenerate states so that conditions (i) and (ii) apply [see Fig. 2e and section 3 in the "Methods"].

**Crossover to clean-limit optical responses.** In real materials, the disorder is present even in very clean samples. We thus need to compare the magnitude of the disorder-mediated and intrinsic responses to characterize the detectability of the latter. We estimate the magnitude of the intrinsic response by counting the dimension of the conductivity tensor in Eq. (3), which gives $\sigma_{\text{int}}(\omega) \simeq \frac{e^2}{h} \frac{1}{\hbar\omega} \frac{(2\Delta)^2}{E_F} k_F^{d-2} \alpha^2$ above the gap, where $k_F$ and $E_F$ are the Fermi wave number and Fermi energy. Here, $0 \leq \alpha \leq 1$ is the ratio between the dominant pairing $\Delta$ and the pairing that are responsible for the optical conductivity. Comparing this with the disorder-mediated response, we obtain

$$\frac{\sigma_{\text{int}}(\omega)}{\sigma_{\text{dis}}(\omega)} \simeq \frac{\omega}{2\Delta} \frac{l}{\xi_0} \left(\frac{2\Delta}{E_F}\right)^2 \alpha^2 \qquad (6)$$

above the superconducting gap, where we use that $\sigma_{\text{dis}}(\omega) \approx \sigma_n(\omega)$[5,7] as shown in Fig. 1d, where $\sigma_n$ is the Drude conductivity in the normal state (see section 5 in the Methods). We thus find that $\sigma_{\text{int}} \gtrsim \sigma_{\text{dis}}$ at $\hbar\omega \sim 2\Delta$ when

$$x \equiv l/\xi_0 > x_c = (k_F\xi_0)^2 \alpha^{-2}. \qquad (7)$$

Since $x_c \gg 1$ in general because $k_F\xi_0 \sim E_F/\Delta \gg 1$ ($E_F/\Delta$ is about $10^4$ for pure metals and $10^2$ for most unconventional superconductors), the clean limit for optical responses is realized in samples much cleaner than that are usually thought to be clean,

**Table 2 Properties of time-reversal-symmetric constant pairing functions in a 2D model of FeSe at $\Gamma$.**

| $\Delta$ | Matrix | $m_x$ | $m_y$ | $m_z$ | $c_{4z}$ | Node | Lowest excitation |
|---|---|---|---|---|---|---|---|
| $\tilde{\Delta}_1$ | $i\sigma_y$ | + | + | + | + | Gapped | No |
| $\tilde{\Delta}_2$ | $\rho_z i\sigma_y$ | + | + | + | − | Gapped | Yes |
| $\tilde{\Delta}_3$ | $\rho_y\sigma_z i\sigma_y$ | + | + | + | + | Gapped | Yes |
| $\tilde{\Delta}_{4a}$ | $\rho_y\sigma_x i\sigma_y$ | − | + | − | $\tilde{\Delta}_{4b}$ | Line | Yes |
| $\tilde{\Delta}_{4b}$ | $\rho_y\sigma_y i\sigma_y$ | + | − | − | $-\tilde{\Delta}_{4a}$ | Line | Yes |
| $\tilde{\Delta}_5$ | $\rho_x i\sigma_y$ | − | − | + | − | Point | Yes |

Here, $\tilde{\Delta}_1 = \Delta_1 i\sigma_y$, $\tilde{\Delta}_2 = \Delta_2\rho_z i\sigma_y$, $\tilde{\Delta}_3 = \Delta_3\rho_y\sigma_z i\sigma_y$, $\tilde{\Delta}_4 a = \Delta_{4a}\rho_y\sigma_x i\sigma_y$, $\tilde{\Delta}_{4b} = \Delta_{4b}\rho_y\sigma_y i\sigma_y$, and $\tilde{\Delta}_5 = \Delta_5\rho_x i\sigma_y$. The second column shows the result of transformation $u_g \Delta u_g^T$ by $g = m_x$, $m_y$, $m_z$ or $c_{4z}$. The signs + and − mean $+\Delta$ and $-\Delta$, respectively.

just satisfying $x > 1$. This explains how the Mattis–Bardeen-type theories have successfully calculated optical conductivity even in clean superconductors.

**Application to FeSe.** In FeSe, however, intrinsic optical responses can make up a significant portion of the observed signal in far-infrared optical measurements. FeSe is a clean quasi-two-dimensional material that has a remarkably large ratio $\Delta/E_F \gtrsim 0.1$[37] with significant spin–orbit coupling comparable to the Fermi energy[38] and strongly orbital-dependent pairing[14], such that $\alpha \sim 1$ is expected. It, therefore, satisfies all the requirements for significant intrinsic optical responses. Here, we use the low-energy model of FeSe in ref. [39] to demonstrate our theory, focusing on the Fermi surface near $\Gamma = (0, 0)$ for simplicity (see section 6 in the "Methods").

We consider six constant pairing functions $\Delta_1$, $\Delta_2$, $\Delta_3$, $\Delta_{4a}$, $\Delta_{4b}$, and $\Delta_5$ that preserve time-reversal symmetry whose matrix forms and symmetries are given in the "Methods" and Table 2. All of

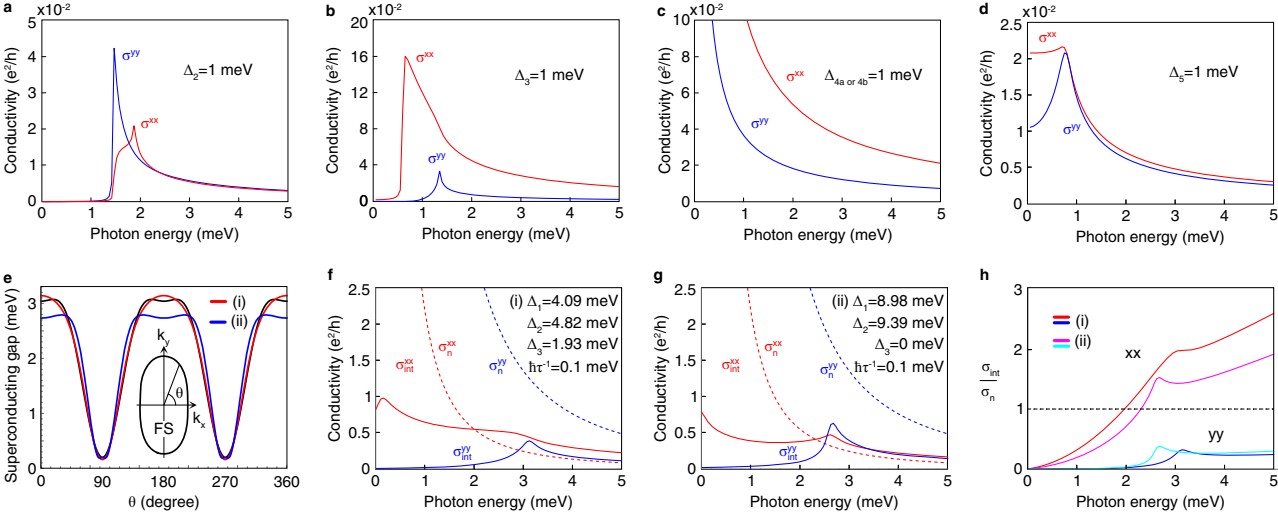

**Fig. 3 Optical conductivity in a model of superconducting FeSe near Γ at zero temperature.** Model parameters in the normal state are adapted from ref. [39] (see section 6 in the Methods). The $xx$ and $yy$ components of the conductivity tensor is shown in red and blue, respectively, in (**a-d**, **f**, **g**). **a-d** Nonzero intrinsic optical conductivity tensors for each constant pairing function. See Methods and Table [2] for the matrix form and symmetries of the six constant pairing functions $\Delta_1$-$\Delta_{4a}$, $\Delta_{4b}$, and $\Delta_5$. The case of the $\Delta_1$ pairing is not shown as the conductivity is identically zero. **e** Superconducting gap similar to the experimentally observed gap. FS and the ellipse enclosing it represent the Fermi surface. Red and blue curves correspond to the choice of pairing functions (i) $\Delta_1 = 4.09$ meV, $\Delta_2 = 4.82$ meV, and $\Delta_3 = 1.93$ meV or (ii) $\Delta_1 = 8.98$ meV, $\Delta_2 = 9.39$ meV, and $\Delta_3 = 0$ meV, respectively. They are least-square fits with and without $\Delta_3$ to $\Delta(\theta) = |2.06 + 1.42\cos(2\theta) - 0.44\cos(4\theta)|$ (shown as a black curve) that was obtained in ref. [15] from experimental data. **f**, **g** Conductivity with pairing functions used in (**e**). $\sigma_{int}$ is the internal optical conductivity in the superconducting state (solid lines), and $\sigma_n$ is the Drude conductivity in the normal state (dashed lines). The disorder-mediated conductivity in the superconducting state is expected to be comparable to $\sigma_n$. **h** Ratio of $\sigma_{int}$ and $\sigma_n$. Red and blue curves are for parameters in (**f**), and magenta and cyan are for parameters in (**g**). $xx$ and $yy$ indicate the component of the conductivity tensor. $\sigma^{xy}$ and $\sigma^{yx}$ are not shown in all plots because they vanish due to $M_x$ symmetry.

them have even parity, but spin-triplet pairing can occur due to their multi-orbital nature ($\Delta_3$ and $\Delta_{4a,4b}$). As we show in the "Methods" section 7, for even parity pairing, optical transitions are not forbidden within each $M_z$ eigenspace in spin–orbit coupled systems. For $\Delta_{i=1,2,3,5}$ pairing, $\mathfrak{C}$ symmetry does not exist within a mirror sector. On the other hand, for $\Delta_{i=4a,4b}$, $\mathfrak{C}$ symmetry exists but satisfies $\mathfrak{C}^2 = 1$ in each mirror sector. In accordance with our analysis, all multi-band pairing $\Delta_{i=2,3,4a,4b,5}$ allow for non-zero optical responses [Fig. 3a–d]. In the case of $\Delta_{i=4a,4b}$, and $\Delta_5$ pairing, optical conductivity tensors are non-zero down to zero frequency because of their gapless spectrum due to the BFS and Dirac points, respectively [Fig. 3d, e].

In experiments, a highly anisotropic pairing gap was observed[14,15], having a sinusoidal shape with 2–3 meV peak at $k_y = 0$ and almost zero deep at $k_x = 0$. Supposing that the gap function belongs to the trivial representation of the symmetry group, we can obtain a similar anisotropic gap with various combinations of $\Delta_1$, $\Delta_2$, and $\Delta_3$. For example, we obtain (i) $\Delta_1 = 4.09$ meV, $\Delta_2 = 4.82$ meV, and $\Delta_3 = 1.93$ meV and (ii) $\Delta_1 = 8.98$ meV, $\Delta_2 = 9.39$ meV, and $\Delta_3 = 0$ meV, respectively, by least-square-fitting with and without spin–orbit coupled pairing $\Delta_3$ to the function $\Delta(\theta) = |2.06 + 1.42\cos(2\theta) - 0.44\cos(4\theta)|$ derived from experimental data[15]. See Fig. 3e. Despite the huge difference in pairing functions for (i) and (ii), the obtained conductivity is quite similar, as shown in Fig. 3f–h. When $\hbar\tau^{-1} = 0.1$ meV ($l/\xi_0 \sim 10^2$), the $xx$ component of the intrinsic optical conductivity in the superconducting state $\sigma_{int}^{xx}$ exceeds the Drude conductivity in the normal state $\sigma_n^{xx}$ at around $\hbar\omega \sim 2$ meV in both cases. Since the disorder-mediated response in the superconducting state is comparable to the normal-state response $\sigma_n$, the intrinsic response dominates above $\hbar\omega \sim 2$ meV along the $x$ direction.

## Discussion
Our theory establishes the existence of the true clean-limit optical responses beyond the Mattis–Bardeen theory in multiband

superconductors. While we focus on linear responses in the current work, our classification of optical transitions applies to nonlinear optical responses also[30]. Since nonlinear optical conductivity tensors have more components than the linear counterpart, they give richer information on the symmetry of the system. For instance, it is hard to detect inversion symmetry breaking from linear optical responses. On the other hand, since second-order optical responses are allowed only when inversion symmetry is broken, they directly reveal the presence of inversion symmetry[30]. As such, various optical measurements can be used in the study of clean multiband superconductors. We anticipate an immediate impact of our work on the optical study of the exotic superconductivity in FeSe. Furthermore, our theory may be relevant to the recently discovered 2D superconductivities reaching $\Delta/E_F > 0.1$ in twisted trilayer graphene[40,41] and ZrNCl[42]. As the synthesis of extremely clean superconductors advances further, our results will become relevant to more materials.

## Methods
**Selection rule by $\mathfrak{C}$ symmetry.** Equation (5) can be simply derived as follows.

$$\langle \mathfrak{C} \cdot nk|V^a(\mathbf{k})nk\rangle = \langle \mathfrak{C}V^a(\mathbf{k}) \cdot nk|\mathfrak{C}^2 \cdot nk\rangle$$
$$= \mathfrak{C}^2\langle \mathfrak{C} \cdot nk|\mathfrak{C}V^a(\mathbf{k})\mathfrak{C}^{-1}|nk\rangle \qquad (8)$$
$$= \epsilon_{\mathfrak{C},V}\mathfrak{C}^2\langle \mathfrak{C} \cdot nk|V^a(\mathbf{k})|nk\rangle.$$

We use that $\mathfrak{C}$ is a anti-unitary operator in the first line, use that $\mathfrak{C}^2 = \pm 1$ is a number, and $V^a$ is Hermitian in the second line, and define $\epsilon_{\mathfrak{C},V} = \pm 1$ by $\mathfrak{C}V^a(\mathbf{k})\mathfrak{C}^{-1} = \epsilon_{\mathfrak{C},V}V^a(\mathbf{k})$ in the third line. Let us recall that $V^a(\mathbf{k}) = \tau_z\partial_{k_a}H(\mathbf{k})$ for a BdG Hamiltonian $H(\mathbf{k})$. $\mathfrak{C}$ anti-commutes with $\tau_z$ because $C$ anti-commutes with $\tau_z$ while a physical unitary operator $U_g$ that combine to define $\mathfrak{C} = U_gC$ commutes with $\tau_z$. Since the $\mathfrak{C}$ symmetry condition imposes $\mathfrak{C}H(\mathbf{k})\mathfrak{C}^{-1} = -H(\mathbf{k})$, we obtain $\epsilon_{\mathfrak{C},V} = 1$, i.e., $\mathfrak{C}$ commutes with $V^a(\mathbf{k})$. Equation (5) then follows.

We note that normal-state systems with an emergent $\mathfrak{C}$ symmetry follow a different selection rule because they satisfy $\epsilon_{\mathfrak{C},v} = -1$. The difference comes from the fact that, in the normal state, all quasi-particles are electronic quasi-particles that couple to the gauge field with the equal charge $-e$ (i.e., the emergent $\mathfrak{C}$ does not reverse the gauging charge). In this case, the velocity operator is $v^a(\mathbf{k}) = \partial_{k_a}h(\mathbf{k})$,

where $h$ is the $\mathbb{C}$-symmetric normal-state Hamiltonian, so that $\mathbb{C}v^a(\mathbf{k})\mathbb{C}^{-1} = \partial_{k_a}\mathbb{C}h(\mathbf{k})\mathbb{C}^{-1} = -v^a(\mathbf{k})$. The selection rule is then $\langle \mathbb{C}\cdot n\mathbf{k}|v^a(\mathbf{k})|n\mathbf{k}\rangle = 0$ when $\mathbb{C}^2 = +1$, which is opposite to the superconducting case.

**Optical excitations with a single-band condition.** Let us suppose that the normal state is described by a single band, i.e., $h(\mathbf{k}) = \xi(\mathbf{k})$ is a $1 \times 1$ matrix. We consider the normal state having time-reversal symmetry or inversion symmetry (or other symmetries whose action is equivalent to them[43]) because only then pairing between two electrons $c_\mathbf{k}$ and $c_{-\mathbf{k}}$ effectively occurs at the Fermi level. Then, $\xi(\mathbf{k}) = \xi(-\mathbf{k})$ such that $V^a(\mathbf{k}) = \hbar^{-1}\partial_a\xi(\mathbf{k})\tau_0$ has vanishing inter-band matrix components for all $\mathbf{k}$, where $\tau_0$ is the $2 \times 2$ identity matrix with the particle–hole indices. It follows that superconductivity in a single-band metal cannot exhibit nontrivial optical conductivity in the clean limit. The same is true when the band has spin degeneracy at every $\mathbf{k}$, because $h(\mathbf{k}) = \xi(\mathbf{k})\sigma_0$ is again proportional to the identity matrix such that the velocity operator is diagonal.

These constraints can be understood from the $\mathbb{C}$ symmetry. Let us note that no identity term appears in the BdG Hamiltonian because we consider $\xi(\mathbf{k}) = \xi(-\mathbf{k})$. Thus, general two-band BdG Hamiltonian takes the form $H = g_x\tau_x + g_y\tau_y + g_z\tau_z$. It always satisfies $\mathbb{C}H(\mathbf{k})\mathbb{C}^{-1} = -H(\mathbf{k})$ for $\mathbb{C} = i\tau_y K$, which satisfies $\mathbb{C}^2 = -1$. This symmetry blocks optical transitions by Eq. (5). Let us consider the case where bands are twofold degenerate due to $\mathfrak{T}^2 = -1$ symmetry of the BdG Hamiltonian, where $\sigma_y$ is an effective-spin Pauli matrix. As we assume time reversal or inversion symmetry, we also have $\mathbb{C}$ and $S$ symmetries. We consider two cases with $\mathbb{C}^2 = 1$ and $\mathbb{C}^2 = -1$ by taking $\{\mathfrak{T} = i\sigma_y K, \mathbb{C} = \tau_x K, S = \tau_x\sigma_y\}$ and $\{\mathfrak{T} = i\tau_z\sigma_y K, \mathbb{C} = i\tau_y K, S = \tau_x\sigma_y\}$, respectively, such that $H = \xi\tau_z\sigma_0 + \Delta_s\tau_y\sigma_y$, and $H = \xi\tau_z\sigma_0 + (\Delta_t \cdot \sigma_y i\sigma_y\tau_+ + h.c.)$, where $\tau_+ = (\tau_x + i\tau_y)/2$. Since these correspond to the spin-singlet and spin-triplet pairing, there is a continuous spin rotation symmetry around an axis, such that the spin around the axis is a good quantum number. Each spin sector is thus described by a two-band BdG Hamiltonian having a $\mathbb{C}$ symmetry $\mathbb{C}^2 = -1$.

We can extend the above results to show that the lowest-energy excitations, from $-E$ to $E$, are forbidden when a multi-band system satisfies the zero superconducting fitness[20,44] condition, i.e., $[h(\mathbf{k}), \Delta^*(-\mathbf{k})] = 0$, which always holds when $\Delta(\mathbf{k})$ or $h(\mathbf{k})$ is proportional to the identity matrix. After we take simultaneous eigenstates of $h(\mathbf{k})$ and $\Delta^*(-\mathbf{k})$, the BdG Hamiltonian decomposes into a set of $2 \times 2$ blocks ($4 \times 4$ blocks, in the presence of spin degeneracy), each of which correspond to a single-band superconductivity. It follows that transitions between two states with energies $-E$ and $E$ are forbidden.

Let us explain it in more detail. When the zero superconducting fitness condition is satisfied, one can take eigenstates $|\alpha\mathbf{k}\rangle$ that satisfy $h(\mathbf{k})|\alpha\mathbf{k}\rangle = \xi_\alpha(\mathbf{k})|\alpha\mathbf{k}\rangle$, $\Delta^*(-\mathbf{k})|\alpha\mathbf{k}\rangle = \Delta_\alpha^*(-\mathbf{k})|\alpha\mathbf{k}\rangle$. In this basis, the BdG Hamiltonian is diagonalized into blocks labeled by $\alpha$:

$$H_\alpha(\mathbf{k}) = \begin{pmatrix} \xi_\alpha(\mathbf{k}) & \Delta_\alpha(\mathbf{k}) \\ -\Delta_\alpha^*(-\mathbf{k}) & -\xi_\alpha(-\mathbf{k}) \end{pmatrix} \quad (9)$$

where the basis states $(1\,0)^T$ and $(0\,1)^T$ correspond to $|\alpha\mathbf{k}\rangle$ and $|\alpha-\mathbf{k}\rangle^*$, respectively. We assume either time reversal symmetry or inversion symmetry of the normal states, such that $\xi_\alpha(\mathbf{k}) = \xi_\alpha(-\mathbf{k})$.

The energy eigenstates of the BdG Hamiltonian are then

$$\begin{aligned} |\alpha, +, \mathbf{k}\rangle &= \begin{pmatrix} \cos\theta_\alpha(\mathbf{k})|\alpha\mathbf{k}\rangle \\ \sin\theta_\alpha(\mathbf{k})|\alpha-\mathbf{k}\rangle^* \end{pmatrix}, \\ |\alpha, -, \mathbf{k}\rangle &= \begin{pmatrix} -\sin\theta_\alpha(\mathbf{k})|\alpha\mathbf{k}\rangle \\ \cos\theta_\alpha(\mathbf{k})|\alpha-\mathbf{k}\rangle^* \end{pmatrix}, \end{aligned} \quad (10)$$

where $\cos\theta_\alpha(\mathbf{k}) = \xi_\alpha(\mathbf{k})/E_{\alpha,+}(\mathbf{k})$, $\sin\theta_\alpha(\mathbf{k}) = \Delta_\alpha(\mathbf{k})/E_{\alpha,+}(\mathbf{k})$, and $E_{\alpha,\pm}(\mathbf{k}) = \pm\sqrt{\xi_\alpha^2(\mathbf{k}) + \Delta_\alpha^2(\mathbf{k})}$. We have

$$\begin{aligned} \langle \alpha, +, \mathbf{k}|v^a|\beta, -, \mathbf{k}\rangle &= \cos\theta(\mathbf{k})\sin\theta(\mathbf{k})\left[v_{\alpha\beta}^a(\mathbf{k}) + v_{\beta\alpha}^a(-\mathbf{k})\right] \\ &= 0 \quad (\text{for } \alpha = \beta), \end{aligned} \quad (11)$$

where the normal-state velocity operator

$$v_{\alpha\beta}^a(\mathbf{k}) = \hbar^{-1}\langle\alpha\mathbf{k}|\partial_a h(\mathbf{k})|\beta\mathbf{k}\rangle \quad (12)$$

satisfy $v_{\alpha\alpha}^a(\mathbf{k}) = -v_{\alpha\alpha}^a(-\mathbf{k})$ due to either time-reversal or inversion symmetry. Thus, assuming nondegenerate states, we immediately see that all transitions from $E_{\alpha,-}(\mathbf{k})$ to $E_{\alpha,+}(\mathbf{k}) = -E_{\alpha,-}(\mathbf{k})$ are forbidden because of the vanishing velocity matrix elements.

The transition between two states with energies $-E$ and $E$ is forbidden even when spin (or pseudo-spin in spin–orbit coupled systems) degeneracy exists. Spin degeneracy can be imposed either by PT symmetry or spin rotation symmetry. In either case, the normal-state velocity matrix element between two spin-degenerate states $|\alpha\mathbf{k}\rangle$ and $|\beta\mathbf{k}\rangle$ vanishes by symmetry, i.e., $v_{\alpha\beta}^a(\mathbf{k}) = 0$. Let us first consider the case where the degeneracy is due to PT symmetry satisfying $(\text{PT})^2 = -1$. Then we have $v_{\alpha,\beta=\text{PT}\alpha}^a(\mathbf{k}) = -v_{\alpha,\text{PT}\alpha}^a(\mathbf{k})$. When spin rotation symmetry exists, the

constraint

$$v_{\uparrow\downarrow}^a(\mathbf{k}) = 0 \quad (13)$$

simply follows from the spin-rotation invariance of the velocity operator. Equation (11) then shows $\langle\alpha, +, \mathbf{k}|v^a|\beta, -, \mathbf{k}\rangle = 0$ for $\alpha$ and $\beta$ related by either PT or a spin rotation. Combining this with $\langle\alpha, +, \mathbf{k}|v^a|\alpha, -, \mathbf{k}\rangle = 0$, we see that all transition channels from $E_{\alpha,-} = E_{\beta,-}$ to $-E_{\alpha,-} = -E_{\beta,-}$ are forbidden.

**Optical excitations with spin rotation symmetries.** Let us consider the EAZ classes within spin sectors for example. We first assume that time reversal, spatial inversion, a spin U(1) rotation and a spin $\pi$ rotation (around an axis perpendicular to the U(1) axis) symmetries are all present. Let us take energy eigenstates such that they carry a definite spin along the $z$ direction. In the case of triplet pairing, this means that we take the *z-direction* as the triplet spin direction because continuous spin rotation symmetries around other directions are broken. Within each spin sector, bands are nondegenerate at generic momenta. As we show in Fig. 2d, PC flips the spin because of particles-hole conjugation. However, the combination of spin $\pi$ rotation, which is $-i\sigma_y$ for singlet pairing ($-i\tau_z\sigma_y$ for triplet pairing), and PC acts within a spin sector, so symmetry under this $\mathbb{C}$-type operation constrains optical excitations through Eq. (5). Optical excitations between $\mathbb{C}$-related pairs are allowed when $\mathbb{C}^2 = (-i\sigma_y\text{PC})^2 = -(\text{PC})^2 = 1$ and $\mathbb{C}^2 = (-i\tau_z\sigma_y\text{PC})^2 = (\text{PC})^2 = 1$, respectively, for singlet and triplet pairing, where BFSs are stable. Since $(\text{PC})^2 = +1$ $(-1)$ for even- and odd-parity pairing (see section 4 in the "Methods" below), this requires exotic odd-parity singlet pairing or even-parity triplet pairing, which is possible only when multi-orbital pairing, e.g., an orbital triplet, is realized. Alternatively, if inversion symmetry is broken or spin–orbit coupling is not negligible, optical excitations are allowed with fully gapped superconductivity. Let us note that $s_z$-preserving spin-orbit coupling is enough to allow optical excitations, because it breaks spin rotation symmetries around other axes such that $\mathbb{C} \propto i\sigma_y\text{PC}$ is broken in each spin sector. In general, spin–orbit coupling breaks all spin rotation symmetries, so two excitation channels are allowed [Fig. 2e].

**Symmetry operator and pairing symmetry.** Let $u_g$ be a unitary operator that acts on space as $g:\mathbf{k} \to g\mathbf{k}$. Suppose that it is a symmetry operator of the normal state, i.e.

$$u_g h(\mathbf{k}) u_g^{-1} = h(g\mathbf{k}), \quad (14)$$

and the pairing function has eigenvalues $e^{i\theta_g}$ under $U_g$, i.e.

$$u_g \Delta(\mathbf{k}) u_g^T = e^{i\theta_g} \Delta(g\mathbf{k}). \quad (15)$$

Due to the non-trivial symmetry transformation of the pairing function, the BdG Hamiltonian is symmetric under

$$U_g = \begin{pmatrix} u_g & 0 \\ 0 & e^{i\theta_g}u_g^* \end{pmatrix}, \quad (16)$$

which rotates the hole sector by $e^{i\theta_g}$ more

$$U_g H(\mathbf{k}) U_g^{-1} = H(g\mathbf{k}). \quad (17)$$

$U_g$ satisfies the following commutation relation with the particle–hole conjugation operator $C$

$$U_g C = e^{i\theta_g} C U_g. \quad (18)$$

Let us take two examples.

1. $U_g = P$ is spatial inversion: $e^{i\theta_g} = +1$ and $-1$ indicates even-parity and odd-parity pairing. Thus, PC = +CP (PC = −CP) for even-parity (odd-parity) pairing.
2. $U_g$ is a spin rotation around the $y$-axis by $\pi$: $e^{i\theta_g} = +1$ always for a spin-singlet pairing, and $e^{i\theta_g} = +1$ $(-1)$ when the pairing function is a spin-triplet with its spin parallel (perpendicular) to the $y$-axis.

**Estimates of disorder-mediated and intrinsic responses in the clean regime.** The disorder-mediated response in the superconducting state is comparable to the Drude response in the normal state. When the light frequency $\omega$ is much larger than the inverse relaxation time $\Gamma$, which is the case in the clean regime $\hbar\Gamma \ll \Delta \lesssim \hbar\omega$, the Drude conductivity is $\sigma_n(\omega) = \sigma_0\frac{\Gamma^2}{\omega^2+\Gamma^2} \approx \sigma_0\frac{\Gamma^2}{\omega^2}$. Here, $\sigma_0 = \frac{ne^2\tau}{m} \approx \frac{e^2}{\hbar}k_F^{d-2}(\hbar^{-1}E_F\tau)$ is the DC conductivity, where $m = \hbar^2 k_F^2/(2E_F)$, and $n \sim k_F^d$. Since $\sigma_{\text{dis}} \sim \sigma_n$, we have

$$\sigma_{\text{dis}}(\omega) \sim \frac{e^2}{h}k_F^{d-2}\frac{E_F}{2\Delta}\left(\frac{\xi_0}{l}\right)\left(\frac{2\Delta}{\hbar\omega}\right)^2, \quad (19)$$

where we use $E_F \sim \hbar v_F k_F$, $\Delta \sim \hbar v_F \xi_0^{-1}$, and $l = v_F\Gamma^{-1}$.

To estimate the intrinsic response. let us note that the inter-band velocity operator is linear in the leading order of $\Delta'/E_F$, where $\Delta'$ is the largest multiband pairing that is allowed to generate interband transitions by the selection rules in Eqs. (4) and (5). The conductivity tensor in Eq. (3) is thus proportional to $\Delta'^2$.

Using that the delta function has the dimension $\omega^{-1}$, and using the Fermi energy and wave number as other scales, we obtain

$$\sigma_{\rm int}(\omega) \sim \frac{e^2}{h}\frac{1}{\hbar\omega}\frac{(2\Delta)^2}{E_F}k_F^{d-2}\alpha^2, \qquad (20)$$

where $\alpha = |\Delta'/\Delta|$ is equal to or smaller than one since $\Delta$ is the dominant pairing strength by definition. It follows that

$$\frac{\sigma_{\rm int}(\omega)}{\sigma_{\rm dis}(\omega)} \sim \frac{\omega}{2\Delta}\frac{l}{\xi_0}\left(\frac{2\Delta}{E_F}\right)^2\alpha^2. \qquad (21)$$

above the superconducting gap frequency.

**FeSe model at the $\Gamma$ point**. If we regard FeSe as a 2D system, it has three Fermi surfaces around $\Gamma = (0,0)$, $X = (\pi, 0)$, and $Y = (0, \pi)$, respectively, in the 1-Fe Brillouin zone[16]. Here, we consider the Fermi surface near $\Gamma$. The Fermi surface of FeSe at $\Gamma$ consists mainly of two orbital degrees of freedom $d_{yz}$ and $d_{xz}$, so we take $\psi(\mathbf{k}) = \left(d_{yz}(\mathbf{k}), d_{xz}(\mathbf{k})\right)^T$ as the basis state. At zero temperature, the normal-state Hamiltonian has the form

$$h_\Gamma = h_0 + h_{\rm nem} + h_{\rm SOC}, \qquad (22)$$

where $h_0 = \epsilon_\Gamma - A(k_x^2 + k_y^2) + B(k_x^2 - k_y^2)\rho_z - 2Ck_xk_y\rho_x$ is the most general spin-less Hamiltonian in the tetragonal phase up to second order in $\mathbf{k}$, $h_{\rm nem} = -D\rho_z$ is the constant part of the nematic terms that develops below $T_{\rm nem} \sim 90$ K[45–47], and $h_{\rm SOC} = \lambda\rho_y\sigma_z$ is the constant spin–orbit coupling. Here, $\rho_{i=x,y,z}$ and $\sigma_{i=x,y,z}$ are the Pauli matrices for orbital and spin degrees of freedom, respectively. $h_0$ and $h_{\rm SOC}$ have tetragonal $D_{4h}$ symmetry under mirror $m_x = -i\rho_z\sigma_x$, $m_y = i\rho_z\sigma_y$, and $m_z = -i\sigma_z$, and fourfold rotation $c_{4z} = i\rho_y e^{-\frac{i\pi}{4}\sigma_z}$, while $h_{\rm nem}$ breaks $c_{4z}$ symmetry down to $c_{2z}$ symmetry. All have time-reversal $t = i\sigma_y K$ symmetry. In our numerical calculations, we take parameters used in ref. [39], which are $\epsilon_\Gamma = -9$ meV, $A = 700$ meVÅ$^2$, $B = C = 484$ meVÅ$^2$, $D = 15$ meV, and $\lambda = 10$ meV.

The bulk FeSe shows superconductivity below 8 K without a sign of time-reversal symmetry breaking[16,45]. We consider constant pairing functions invariant under time reversal. Since there are six $4 \times 4$ matrices that are invariant under $t = i\sigma_y K$, the pairing function has the form

$$\Delta(\mathbf{k}) = \Big(\Delta_1 + \Delta_2\rho_z + \Delta_3\rho_y\sigma_z$$
$$+ \Delta_{4a}\rho_y\sigma_x + \Delta_{4b}\rho_y\sigma_y + \Delta_5\rho_x\Big)i\sigma_y, \qquad (23)$$

where $\Delta_1$, $\Delta_2$, $\Delta_3$, $\Delta_{4a}$, $\Delta_{4b}$, and $\Delta_5$ are all independent of $\mathbf{k}$. The pairing symmetry of each term under the $D_{4h}$ point group is shown in Table 2. Let us note that all are even-parity pairing, i.e., invariant under $p = m_xm_ym_z = 1$. The orbital-singlet nature of $\Delta_3$ and $\Delta_{4a,4b}$ allows them to be spin triplet even though they have even parity.

**Optical excitations in $M_z$, $P$, and $T$-symmetric 2D superconductors**. In two-dimensional systems perpendicular to the $z$ axis, $M_z$ symmetry divides eigenstates into two distinct eigenspaces with $M_z$ eigenvalues $\lambda = \pm i$. It imposes a selection rule.

Let $M_zC = \eta CM_z$, where $\eta = \pm 1$, and $M_z|n\rangle = \lambda_n|n\rangle$. Then, $M_z\text{PC}|n\rangle = \eta\text{PC}M_z|n\rangle = \eta\text{PC}\lambda_n|n\rangle = \eta\lambda_n^*\text{PC}|n\rangle = -\eta\lambda_n\text{PC}|n\rangle$. Thus, $|n\rangle$ and $\text{PC}|n\rangle$ has the different eigenvalues when $\eta = 1$ and has same eigenvalues when $\eta = -1$. In the former case, the optical transition from $|n\rangle$ to $\text{PC}|n\rangle$ is forbidden by $M_z$ symmetry because they are indifferent eigenspaces (and the velocity operator does not change the eigenspace), but the transition from $|n\rangle$ to $S|n\rangle$ is allowed. On the other hand, in the latter case, the transition from $|n\rangle$ to $\text{PC}|n\rangle$ within the same mirror sector is forbidden when the pairing is odd-parity such that $(\text{PC})^2 = -1$.

In summary, optical transitions between particle-hole- and chiral-related states are forbidden in $P$ and $T$-symmetric systems when $M_zC = -CM_z$ and $\text{PC} = -CP$. Similar constraints can appear in one-dimensional systems also due to mirror symmetries.

## Data availability

The data that support the findings of this study are available from the corresponding author upon reasonable request.

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

## Acknowledgements

We appreciate Takasada Shibauchi, Kenichiro Hashimoto, Yuta Mizukami, and Guang-Yu Guo for helpful discussions and thank Maine Christos for their useful comments on the manuscript. J.A. was supported by the RIKEN Special Postdoctoral Researcher Program, the funding via Ashvin Vishwanath from the Center for Advancement of Topological Semimetals, an Energy Frontier Research Center funded by the US Department of Energy Office of Science, Office of Basic Energy Sciences, through the Ames Laboratory under contract No. DE-AC02-07CH11358, and Basic Science Research Program through the National Research Foundation of Korea (NRF) funded by the Ministry of Education (Grant no. 2020R1A6A3A03037129). N.N. was supported by JST CREST Grant nos. JPMJCR1874 and JPMJCR16F1, Japan, and JSPS KAKENHI Grant no. 18H03676.

## Author contributions

J.A. conceived the original idea and performed the theoretical analysis. N.N. supervised the project and noticed the relevance of the theory to FeSe. Both authors discussed the data and wrote the paper.

## Competing interests

The authors declare no competing interests.
