## [Peer Review File · Nature Communications]

REVIEWER COMMENTS

Reviewer #1 (Remarks to the Author):

The manuscript by Ahn and Nagaosa discusses mechanisms for optical excitation in superconductors, which is an old problem that the authors have analyzed using new ideas about symmetries in superconductors from the "tenfold way". The main technical question I have is regarding the third mechanism proposed by the authors for optical excitation in a superconductor, based on spin-orbit coupling. I think the first two have appeared in some form in prior work, but this third mechanism has not, as far as I understand, but might be the most general. I believe the result is probably correct and warrants publication in Nature Communications, given its combination of intrinsic theoretical interest with a specific prediction for FeSe.

Is it possible to quantify how strong the effect induced by spin-orbit coupling will be, in a superconductor with inversion symmetry so that there is no effect without spin-orbit? And are there important aspects of the spin-orbit coupling other than just its overall magnitude? This might help link the two parts of the paper, one more mathematically minded and one rather specific to FeSe, although the authors do make a good effort (e.g., in Figure 2) to give some intuition for theoretical results.

Regarding clarity of writing, I think the abstract would be taken by many readers to suggest that people were not aware that breaking inversion allowed optical excitation at the gap threshold; indeed it is in papers cited later in the manuscript. So I think some rewording there is appropriate. The abstract could also clarify whether FeSe has inversion symmetry, i.e., which of the three classes it falls in. I believe that conventionally it does have inversion symmetry, but there has been some work, at least theoretically, on inducing inversion symmetry breaking via a substrate.

My last question is: is there any way to distinguish in an experiment of this type between the mechanisms (i) and (iii)? In other words, if a material has the relevant kind of symmetry breaking, and then inversion symmetry is broken by a perturbation, is there a way to tell apart the two effects? As with new superconductors it can be controversial whether they break inversion or not, any distinction between the two excitation processes the authors consider would be useful.

There are a few minor errors (de Genne instead of de Gennes) that do not affect the meaning. As said above, I think the paper should probably be published in Nature Communications even if there are not straightforward answers to the questions above, but if the answers are easy and quick, it might help to increase further the readability and impact of the paper.

Reviewer #2 (Remarks to the Author):

In this theoretical work, Ahn and Nagaosa revisit the problem of far-infrared optical response of superconductors, which is conventionally described by the momentum-non-conserving Mattis-Bardeen theory even in the case of very clean superconductors. The manuscript contains several valuable observations and arguments that clarify the unreasonable effectiveness of this theory while also foreshadowing its limitations.

On the one hand, it is shown how the absence of momentum-conserving optical transitions in simple BdG-form models of superconductors is enforced by a selection rule imposed by " \mathcal{C} " (composition of space-inversion and particle-hole symmetry). This then allows the authors to adopt the "effective Altland-Zirnbauer" symmetry classes and to develop a tenfold-way-like characterization of lowest optical excitations in superconductors. It is observed that certain unconventional

superconducting pairings, (in particular a large variety of the multi-orbital ones) provide exceptions from the selection rule, thus (at least theoretically) allowing for a deviation from the Mattis-Bardeen theory via direct or “intrinsic” transitions.

On the other hand, by comparing the relevant scales, the authors also illuminate why that deviation remains extremely small even in the cases that do not adhere to the selection rule, and why the momentum-non-conserving processes dominate the response even in superconductors that are commonly considered to be very clean. Towards the end of the manuscript, the authors discuss an intriguing application to FeSe, where the signature of the intrinsic momentum-conserving transitions could potentially be experimentally observed.

I find this an excellent manuscript in many aspects. Besides enhancing the existing theory of the superconducting optical transitions, the proposed application to FeSe is indeed arguably within the experimental reach. The appearance of this work is very timely and would likely motivate experimental research into materials beyond just FeSe. The presented theoretical arguments are for the most part clear and easy to follow, and the organization of the text into the main text and methods makes the manuscript appropriately structured for Nature Communications. For all these reasons, I am inclined to recommend the work for publication in Nat. Comm. assuming that my remaining clarifying questions below are properly resolved.

1.) Notation. In all honesty, I got confused by the authors’ reasoning at multiple occasions simple because “T” (for time reversal) and “ \mathcal{T} ” (for space-time inversion) look too much alike and are easily confused; similarly for “C” and “ \mathcal{C} ”. I am wondering if a less confusing notation could be adopted.

2.) I am wondering about the minus signs on the right-hand side of Eq. (2). Shouldn’t the reverted “+e” charge for the hole sector induce an additional minus sign in the bottom-right block of the velocity operator?

3.) It is not quite clear to me how spin-rotation symmetries are incorporated in the classification result in Table 1. Note that Fig. 3(c) in Ref. [26] where the EAZ classes have previously been introduced explicitly checks the nature of spin-rotation symmetry before telling the corresponding EAZ class of the superconductor. Could the authors clarify this mismatch? To make a concrete example: Time-reversal-breaking superconductors with U(1) spin-rotation symmetry and with inversion symmetry are labelled as class “A” in [26]. How would the authors of the present work label such class of superconductors?

4.) It is stated below Eq. (5) that “ $\mathcal{C}^2 = +1$ ” implies that the superconductor hosts Bogoliubov Fermi surfaces. While codimensions arguments imply that a generic node would indeed be a BFS, why do they necessarily have to occur? Isn’t it possible to realize a fully gapped SC phase?

5.) The two choices of SC order parameters in Fig. 3e appear too arbitrary. Could the authors include a short Supplementary Note commenting on the fits to the experimental data and on the possible choices of $\Delta_{1,2,3}$?

6.) I find the definition of Δ' (with the apostrophe) below Eq. (19) too vague. Surely a more concrete characterization could be given.

7.) On page 7, the statement that “all other pairings allow for non-zero optical responses” seems to suggest “all except the just discussed $\Delta_{4a,4b}$ ” – but that is not what the authors mean. A less confusing formulation would be appropriate.

8.) Typo on page 8: "...quasi-particles that coupled (\diamond couple) to the gauge field..."

Reviewer #3 (Remarks to the Author):

The manuscript under review addresses the question when optical excitations across the gap are allowed by symmetry in clean superconductors. For conventional superconductors, they are forbidden by symmetry, which has motivated the development of the dirty-limit Mattis-Bardeen theory of the optical response. Since this theory successfully describes many experiments, the question of the intrinsic, clean-limit response has not been studied in depth. However, the manuscript shows that this left a large gap in our picture of superconducting materials. The authors find that under several different, and often realized, conditions, optical excitations across the gap are allowed in clear unconventional superconductors. The availability of ultra-clean samples of unconventional superconductors (e.g., FeSe, which is addressed by the authors) as well as the current shift of theoretical perspective on their symmetry-induced properties mean that the manuscript is highly timely and provides important new insights. It is also mostly well written and the conditions for optical excitations are clearly spelled out, which should motivate experimental efforts.

I recommend publication of the manuscript, in principle, but would like to ask the authors to address a number of points, which are listed below. These concern the presentation and, in a few cases, technical problems, which I expect to be uncritical for the results.

* A general point: already in the abstract, the authors state that optical excitations across the gap "can occur in clean multi-band superconductors." It is not obvious to me that the multi-band character (beyond spin) is actually necessary for the optical response. The authors should clarify this.

* In the abstract, I suggest to remind readers that Mattis-Bardeen theory describes the dirty limit.

* In their review of multi-band superconductors, the authors mention Sr₂RuO₄. Here, they might also cite Phys. Rev. Research 2, 032023(R) (2020).

* The name of "Zirnbauer" is misspelled at least twice. Also, in the context of Ref. 27, the authors might also cite the earlier paper J. Math. Phys. 37, 4986 (1996). "de Gennes" is also misspelled.

* The wording below Eq. (1) is somewhat sloppy; $\hat{c}_{\mathbf{k}}$ is not an "electron" but a spinor annihilation operator and its components are $\hat{c}_{\rho s \mathbf{k}}$. Also in this paragraph, the complex-conjugation operator K should be defined.

* In the first paragraph of "Selection rules", the multiplication of scalars should not be denoted by "*".

* Below Eq. (5), calling \mathcal{C} an emergent symmetry seems misleading to me. This is just charge conjugation multiplied by inversion, I think.

* In Methods section 2, 2nd paragraph, the singlet/triplet Hamiltonians H appear to be incorrect. Since the authors use the standard definition of the Nambu spinor (p. 4), the singlet Hamiltonian should be

$$\begin{aligned} H &= (\text{normal part}) \\ &+ ((0, \Delta_s(\mathbf{k}) i\sigma_y), (-\Delta_s^*(\mathbf{k}) i\sigma_y, 0)) \\ &= \dots - \text{Re}\Delta_s(\mathbf{k}) \tau_y \sigma_y - \text{Im}\Delta_s(\mathbf{k}) \tau_x \sigma_y \end{aligned}$$

The triplet part should also contain the factor $i\sigma_y$. The conclusions drawn from the specific forms of H need to be checked.

* In the next paragraph, with regard to superconducting fitness, the authors might also cite the longer paper Phys. Rev. B 98, 024501 (2018).

* In Methods section 3, the authors first assume spin-rotation symmetry to be present and then talk about triplet pairing. This is a contradiction since triplet pairing involves the d-vector, which necessarily breaks spin-rotation symmetry. In the same paragraph, I do not understand $\mathcal{C}^2 = -(PC)^2$. Is this a typo? Since C is antiunitary I do not think that \mathcal{C} can be defined such that this holds.

* In Methods section 6, the authors should state that ρ_i are Pauli matrices in orbital space.

* In Methods section 7, it is sloppy notation to use the same symbol M_z for both an operator and its eigenvalues.

Response to Review #1

Reviewer #1 (Remarks to the Author):

The manuscript by Ahn and Nagaosa discusses mechanisms for optical excitation in superconductors, which is an old problem that the authors have analyzed using new ideas about symmetries in superconductors from the "tenfold way". The main technical question I have is regarding the third mechanism proposed by the authors for optical excitation in a superconductor, based on spin-orbit coupling. I think the first two have appeared in some form in prior work, but this third mechanism has not, as far as I understand, but might be the most general. I believe the result is probably correct and warrants publication in Nature Communications, given its combination of intrinsic theoretical interest with a specific prediction for FeSe.

Authors' response: We thank the reviewer for this positive evaluation of our work. We answer all the questions below.

Is it possible to quantify how strong the effect induced by spin-orbit coupling will be, in a superconductor with inversion symmetry so that there is no effect without spin-orbit? And are there important aspects of the spin-orbit coupling other than just its overall magnitude? This might help link the two parts of the paper, one more mathematically minded and one rather specific to FeSe, although the authors do make a good effort (e.g., in Figure 2) to give some intuition for theoretical results.

Authors' response: It is possible to estimate the magnitude of the spin-orbit-induced optical conductivity using dimensional analysis. When the spin-orbit coupling is the only source of the nonzero optical transition, the transition amplitude is proportional to the spin-orbit coupling E_{SOC} in the leading order of E_{SOC}/E_F . So, we expect $\sigma \sim (E_{\text{SOC}}/E_F)^2$: $\sigma(\omega) \sim \frac{e^2}{h} \frac{1}{\hbar\omega} \frac{(2\Delta)^2}{E_F} k_F^{d-2} \alpha^2$, where $\alpha \sim \left(\frac{E_{\text{SOC}}}{E_F}\right)^2 \left(\frac{\Delta_{\text{multi}}}{\Delta}\right)^2$, 2Δ is the superconducting gap size, and Δ_{multi} is the multi-band pairing. In FeSe, spin-orbit coupling is significant such that $E_F \sim E_{\text{SOC}} \sim 10$ meV.

As for important aspects of spin-orbit coupling other than the overall magnitude of the conductivity tensor, we do not see a universal feature in the linear optical response. For example, the detailed spectral shape is characterized mainly by the (superconducting) band structure rather than by the presence of spin-orbit coupling.

However, nonlinear optical responses can give additional information on the spin-orbit coupling. In contrast to the linear optical conductivity tensor, which is determined solely by the absorption rate and the band structure, nonlinear conductivity tensors contain other quantities. The impact of spin-orbit coupling on those additional components may be able to be revealed in nonlinear optical responses.

Regarding clarity of writing, I think the abstract would be taken by many readers to suggest that people were not aware that breaking inversion allowed optical excitation at the gap threshold; indeed it is in papers cited later in the manuscript. So I think some rewording there is appropriate. The abstract could also clarify whether FeSe has inversion symmetry, i.e., which of the three classes it falls in. I believe that conventionally it does have inversion symmetry, but there has been some work, at least theoretically, on inducing inversion symmetry breaking via a substrate.

Authors' response: We thank the referee for this suggestion. We have revised the abstract to clarify that the FeSe we study has inversion symmetry. Also, we state in the revised abstract that a recent work showed that clean-limit optical responses are possible in inversion-broken superconductors.

My last question is: is there any way to distinguish in an experiment of this type between the mechanisms (i) and (iii)? In other words, if a material has the relevant kind of symmetry breaking, and then inversion symmetry is broken by a perturbation, is there a way to tell apart the two effects? As with new superconductors it can be controversial whether they break inversion or not, any distinction between the two excitation processes the authors consider would be useful.

Authors' response: There is a good way to detect inversion symmetry breaking in optical measurements. It is hard to see inversion symmetry breaking through linear optical responses. However, second-order optical responses, such as the second-harmonic generation or the bulk photovoltaic effect, are sensitive to the inversion symmetry breaking because they are allowed only when inversion symmetry is broken. The second-order optical measurements have been useful in the determination of inversion symmetry of the normal state (the pseudo-gap phase) of cuprates [See, e.g., Zhao et al., Nat. Phys. 13, 250 (2017), Torre et al., arXiv:2008.06516, Lim et al., arXiv:2011.06755]. The same technique can be used to determine the inversion symmetry in the superconducting state. Case (iii) will show only nontrivial linear optical responses, while (i) show both nontrivial linear and second-order optical responses. We added some comments on inversion symmetry and nonlinear optical responses in the revised conclusion.

There are a few minor errors (de Genne instead of de Gennes) that do not affect the meaning. As said above, I think the paper should probably be published in Nature Communications even if there are not straightforward answers to the questions above, but if the answers are easy and quick, it might help to increase further the readability and impact of the paper.

Authors' response: We thank the reviewer for pointing out the typo. We have corrected them in the revised manuscript. Also, we appreciate the reviewer for recommending our paper for publication. We believe that our revised manuscript has been improved indeed in the course of responding to all of the referee's comments.

Response to Review #2

Reviewer #2 (Remarks to the Author):

In this theoretical work, Ahn and Nagaosa revisit the problem of far-infrared optical response of superconductors, which is conventionally described by the momentum-non-conserving Mattis-Bardeen theory even in the case of very clean superconductors. The manuscript contains several valuable observations and arguments that clarify the unreasonable effectiveness of this theory while also foreshadowing its limitations.

On the one hand, it is shown how the absence of momentum-conserving optical transitions in simple BdG-form models of superconductors is enforced by a selection rule imposed by “ \mathcal{C} ” (composition of space-inversion and particle-hole symmetry). This then allows the authors to adopt the “effective Altland-Zirnbauer” symmetry classes and to develop a tenfold-way-like characterization of lowest optical excitations in superconductors. It is observed that certain unconventional superconducting pairings, (in particular a large variety of the multi-orbital ones) provide exceptions from the selection rule, thus (at least theoretically) allowing for a deviation from the Mattis-Bardeen theory via direct or “intrinsic” transitions.

On the other hand, by comparing the relevant scales, the authors also illuminate why that deviation remains extremely small even in the cases that do not adhere to the selection rule, and why the momentum-non-conserving processes dominate the response even in superconductors that are commonly considered to be very clean. Towards the end of the manuscript, the authors discuss an intriguing application to FeSe, where the signature of the intrinsic momentum-conserving transitions could potentially be experimentally observed.

I find this an excellent manuscript in many aspects. Besides enhancing the existing theory of the superconducting optical transitions, the proposed application to FeSe is indeed arguably within the experimental reach. The appearance of this work is very timely and would likely motivate experimental research into materials beyond just FeSe. The presented theoretical arguments are for the most part clear and easy to follow, and the organization of the text into the main text and methods makes the manuscript appropriately structured for Nature Communications. For all these reasons, I am inclined to recommend the work for publication in Nat. Comm. assuming that my remaining clarifying questions below are properly resolved.

Authors’ response: We thank the reviewer for this high evaluation of our work. We also thank the referee for valuable comments and questions. We answer all of them point by point below.

1.) Notation. In all honesty, I got confused by the authors’ reasoning at multiple occasions simple because “T” (for time reversal) and “ \mathcal{T} ” (for space-time inversion) look too much alike and are easily confused; similarly for “C” and “ \mathcal{C} ”. I am wondering if a less confusing notation could be adopted.

Authors' response: We agree that our previous notations may be confusing. In the revised manuscript, we use the fraktur font instead of the mathcal font. Now \mathfrak{T} and \mathfrak{C} are clearly distinguished from time reversal T and particle-hole conjugation C .

2.) I am wondering about the minus signs on the right-hand side of Eq. (2). Shouldn't the reverted "+e" charge for the hole sector induce an additional minus sign in the bottom-right block of the velocity operator?

Authors' response: We thank the referee for pointing out this typo. We have corrected the sign by writing it as $\partial_{k_a}[h^T(-\mathbf{k})]$ in the revised manuscript.

3.) It is not quite clear to me how spin-rotation symmetries are incorporated in the classification result in Table 1. Note that Fig. 3(c) in Ref. [26] where the EAZ classes have previously been introduced explicitly checks the nature of spin-rotation symmetry before telling the corresponding EAZ class of the superconductor. Could the authors clarify this mismatch? To make a concrete example: Time-reversal-breaking superconductors with $U(1)$ spin-rotation symmetry and with inversion symmetry are labelled as class "A" in [26]. How would the authors of the present work label such class of superconductors?

Authors' response: No discrepancy exists between the classification by Bzdusek and Sigrist and the classification by ours. They classify superconducting symmetries into three types: $U(1)$, $SU(2)$, and no spin rotation symmetry. We also consider three cases: $U(1)$, $U(1)$, and one spin- π rotation, and no spin rotation symmetry. Our second case is equivalent to the case with $SU(2)$ spin rotation symmetry as long as EAZ classes are concerned.

For example, consider $U(1)$ spin rotation symmetry and inversion symmetry with broken time reversal symmetry. Let us first forget about the spin rotation symmetry. Then the only symmetry operator relevant for the EAZ classification is PC , the combination of inversion P and particle-hole conjugation C . This operation flips the spin sector because particle-hole conjugation flips the spin while inversion does not. Therefore, if we look at one spin sector (either up or down), which is well-defined due to the spin $U(1)$ rotation symmetry, the PC symmetry is absent. The EAZ class of one spin sector is therefore class A. This is what we explain in Method section 3, and it is consistent with Bzdusek and Sigrist's results.

As another example, we can consider a time-reversal-breaking even-parity pairing with $SU(2)$ spin rotation symmetry and inversion symmetry. Compared to the spin- $U(1)$ case, we have an additional π spin rotation symmetry around the y axis, if we take the spin $U(1)$ axis as the z axis. Since the π rotation symmetry flips the spin z component, PC times the π rotation leaves spin z invariant. So, each spin- z space has an effective \mathfrak{C} symmetry, which is PC times the π rotation. For even-parity pairing, $(PC)^2 = 1$. As a spin π rotation squares to -1 , the \mathfrak{C} squares to -1 , defining the EAZ class C. We explain this in Method 3, and it is again consistent with the result by Bzdusek and Sigrist.

4.) It is stated below Eq. (5) that " $\mathfrak{C}^2=+1$ " implies that the superconductor hosts

Bogoliubov Fermi surfaces. While codimensions arguments imply that a generic node would indeed be a BFS, why do they necessarily have to occur? Isn't it possible to realize a fully gapped SC phase?

Authors' response: We thank the referee for pointing this out. There is no reason why the BFSs must appear, when there is no other symmetry. It is possible to realize a fully gapped phase when $C^2=+1$. We have revised the abstract and the main text such that the symmetry protection of the BFSs, rather than the emergence of the BFSs, is stated as the condition for nontrivial optical excitations. The sentence “ *$C^2=+1$ implies that the superconductor hosts Bogoliubov Fermi surfaces*” now reads “... may host stable BFSs.”.

5.) The two choices of SC order parameters in Fig. 3e appear too arbitrary. Could the authors include a short Supplementary Note commenting on the fits to the experimental data and on the possible choices of $\Delta_{\{1,2,3\}}$?

Authors' response: We agree that our previous superconducting order parameters were not so systematically chosen. In the revised manuscript, we use new two sets of parameters that are chosen by the least-square fitting to experimental data (more precisely, we least-square-fitted with our superconducting parameters to a gap function obtained in Liu et al., PRX 8, 031033 (2018)). There is no qualitative change in the shape of the new gap functions and the conductivity spectra.

Now all the fitting procedure is explained in the main text. We thus believe that no additional Supplemental Note is needed.

6.) I find the definition of Δ' (with the apostrophe) below Eq. (19) too vague. Surely a more concrete characterization could be given.

Authors' response: To make the definition Δ' more concrete, we have revised the relevant sentence as “ *Δ' is the largest multi-band pairing that is allowed to generate inter-band transitions by the selection rules in equations (4) and (5).*”.

7.) On page 7, the statement that “all other pairings allow for non-zero optical responses” seems to suggest “all except the just discussed $\Delta_{\{4a,4b\}}$ ” – but that is not what the authors mean. A less confusing formulation would be appropriate.

Authors' response: We have rewritten that part into “... all multi-band pairing $\Delta_{i=2,3,4a,4b,5}$ allow for non-zero optical responses.” to make it clearer.

8.) Typo on page 8: “...quasi-particles that coupled (\diamond couple) to the gauge field...”

Authors' response: We have corrected “coupled” to “couple” in the revised manuscript.

Response to Review #3

Reviewer #3 (Remarks to the Author):

The manuscript under review addresses the question when optical excitations across the gap are allowed by symmetry in clean superconductors. For conventional superconductors, they are forbidden by symmetry, which has motivated the development of the dirty-limit Mattis-Bardeen theory of the optical response. Since this theory successfully describes many experiments, the question of the intrinsic, clean-limit response has not been studied in depth. However, the manuscript shows that this left a large gap in our picture of superconducting materials. The authors find that under several different, and often realized, conditions, optical excitations across the gap are allowed in clear unconventional superconductors. The availability of ultra-clean samples of unconventional superconductors (e.g., FeSe, which is addressed by the authors) as well as the current shift of theoretical perspective on their symmetry-induced properties mean that the manuscript is highly timely and provides important new insights. It is also mostly well written and the conditions for optical excitations are clearly spelled out, which should motivate experimental efforts.

I recommend publication of the manuscript, in principle, but would like to ask the authors to address a number of points, which are listed below. These concern the presentation and, in a few cases, technical problems, which I expect to be uncritical for the results.

Authors' response: We appreciate the reviewer for recommending our paper for publication. We also thank so much for carefully reading our manuscript including the Methods sections and giving us detailed comments. Those constructive comments were helpful for improving our manuscript. We address all the questions and comments below.

* A general point: already in the abstract, the authors state that optical excitations across the gap "can occur in clean multi-band superconductors." It is not obvious to me that the multi-band character (beyond spin) is actually necessary for the optical response. The authors should clarify this.

Authors' response: The multi-band character is needed because optical transitions in the superconducting state of single-band metals are forbidden by the presence of \mathcal{C} symmetry.

Let us suppose that we construct a single-band metal model describing one band of a metal at the Fermi level. Note that the formation of zero-momentum Cooper pairs requires $\epsilon_{\mathbf{k}} = \epsilon_{-\mathbf{k}}$ in the normal state, time reversal T, inversion P, or any other symmetry that relates \mathbf{k} and $-\mathbf{k}$ can be assumed. In such a case, the 2x2 BdG Hamiltonian has no identity term:

$$H(\mathbf{k}) = \begin{pmatrix} \epsilon_{\mathbf{k}} - \mu & \Delta_{1\mathbf{k}} - i\Delta_{2\mathbf{k}} \\ \Delta_{1\mathbf{k}} + i\Delta_{2\mathbf{k}} & -(\epsilon_{-\mathbf{k}} - \mu) \end{pmatrix} = (\epsilon_{\mathbf{k}} - \mu)\tau_z + \Delta_{1\mathbf{k}}\tau_x + \Delta_{2\mathbf{k}}\tau_y.$$

Thus, it satisfies $i\tau_y K H(\mathbf{k})(i\tau_y K)^{-1} = -H(\mathbf{k})$, showing that $i\tau_y K$ is \mathcal{C} symmetry. Since the \mathcal{C} symmetry, satisfying $\mathcal{C}^2 = -1$, prohibits optical excitations in the superconducting state in single-band models, we draw the conclusion that multi-band effects are necessary for nontrivial optical excitations.

This property was explained in the Methods section 2 of the previous manuscript. In the revised manuscript, we have added more explanations on the \mathcal{C} in single-band models below equation (5) of the main text.

* In the abstract, I suggest to remind readers that Mattis-Bardeen theory describes the dirty limit.

Authors' response: We have added the word “dirty limit” explicitly in the revised abstract.

* In their review of multi-band superconductors, the authors mention Sr2RuO4. Here, they might also cite Phys. Rev. Research 2, 032023(R) (2020).

Authors' response: We have added a citation to Phys. Rev. Research 2, 032023(R) (2020) in the revised introduction.

* The name of "Zirnbauer" is misspelled at least twice. Also, in the context of Ref. 27, the authors might also cite the earlier paper J. Math. Phys. 37, 4986 (1996). "de Gennes" is also misspelled.

Authors' response: We have corrected all the spelling of “Zirnbauer” and “de Gennes” in our revised manuscript. We added citations to the paper J. Math. Phys. 37, 4986 (1996) by Zirnbauer when we cite Altland and Zirnbauer, PRB 55, 1142 (1997).

* The wording below Eq. (1) is somewhat sloppy; $\hat{c}_{\mathbf{k}}$ is not an "electron" but an spinor annihilation operator and its components are $\hat{c}_{\rho s \mathbf{k}}$. Also in this paragraph, the complex-conjugation operator K should be defined.

Authors' response: We have revised the clause below Eq. (1) to

“... in the basis of the Nambu spinor defined by $\hat{\Psi} = (\hat{c}_{\rho s \mathbf{k}}, \hat{c}_{\rho s -\mathbf{k}}^+)^T$, where $\hat{c}_{\rho s \mathbf{k}}$ is the electronic quasiparticle annihilation operator with orbital ρ and spin $s = \uparrow, \downarrow$ indices.”

Also, we state that K is the complex conjugation operator in the revised manuscript.

* In the first paragraph of "Selection rules", the multiplication of scalars should not be denoted by "*".

Authors' response: We have deleted “*” in the revised manuscript.

* Below Eq. (5), calling \mathcal{C} an emergent symmetry seems misleading to me. This is just charge conjugation multiplied by inversion, I think.

Authors' response: We agree with the referee. Since superconducting pairing does not break inversion symmetry in single-band models, \mathcal{C} is identical to the combination of charge conjugation and inversion. We have dropped the term “emergent” in the revised manuscript.

* In Methods section 2, 2nd paragraph, the singlet/triplet Hamiltonians H appear to be incorrect. Since the authors use the standard definition of the Nambu spinor (p. 4), the singlet Hamiltonian should be

$$\begin{aligned} H &= (\text{normal part}) \\ &+ ((0, \Delta_s(\mathbf{k}) i\sigma_y), (-\Delta_s^*(\mathbf{k}) i\sigma_y, 0)) \\ &= \dots - \text{Re}\Delta_s(\mathbf{k}) \tau_y \sigma_y - \text{Im}\Delta_s(\mathbf{k}) \tau_x \sigma_y \end{aligned}$$

The triplet part should also contain the factor $i\sigma_y$. The conclusions drawn from the specific forms of H need to be checked.

Authors' response: We thank the referee for this comment. Our previous expression in that part used a Nambu spinor basis different from the standard basis we used for equation (1). We agree that this may be confusing. In the revised manuscript, we choose the standard basis in the second paragraph of the Method section 2.

* In the next paragraph, with regard to superconducting fitness, the authors might also cite the longer paper Phys. Rev. B 98, 024501 (2018).

Authors' response: We have cited Phys. Rev. B 98, 024501 (2018) also in the revised manuscript.

* In Methods section 3, the authors first assume spin-rotation symmetry to be present and then talk about triplet pairing. This is a contradiction since triplet pairing involves the d-vector, which necessarily breaks spin-rotation symmetry. In the same paragraph, I do not understand $\mathcal{C}^2 = -(\text{PC})^2$. Is this a typo? Since C is antiunitary I do not think that \mathcal{C} can be defined such that this holds.

Authors' response: We thank the referee for these comments. We have clarified the two confusing points in the revised Method section 3 by explaining the followings:

First, the spin rotation symmetry we assume is a spin U(1) rotation symmetry and a spin π rotation around the axis perpendicular to the U(1) axis. In the case of triplet pairing, the U(1) axis is parallel to the \mathbf{d} vector. For concreteness, let us take $\mathbf{d} = |\mathbf{d}|\hat{z}$ and the spin π rotation as $i\sigma_y$. The spin π rotation flips the direction of \mathbf{d} . However, the combination of the spin π rotation and electron phase rotation by 90 degree remains as a symmetry operation. This operator is represented by $i\tau_z\sigma_y$. We called it also a spin π rotation.

Second, \mathcal{C} within a spin-z sector is defined as the product of the spin π rotation ($i\sigma_y$ for singlet and $i\tau_z\sigma_y$ for triplet) and PC , not simply by PC . It is because PC flips the spin-z and the spin π rotation also flips the spin-z. In the singlet pairing case, we have $(\mathcal{C})^2 = (i\sigma_y PC)^2 = -(PC)^2$.

* In Methods section 6, the authors should state that ρ_i are Pauli matrices in orbital space.

Authors' response: We have stated that ρ_i are Pauli matrices in the orbital space in the revised manuscript.

* In Methods section 7, it is sloppy notation to use the same symbol M_z for both an operator and its eigenvalues.

Authors' response: In the revised manuscript, we have substituted M_z with λ when they stand for eigenvalues, not operators.

REVIEWERS' COMMENTS

Reviewer #1 (Remarks to the Author):

The authors have given convincing replies to my questions and I am satisfied to see the paper published. Indeed there seems to be general support for publication from the referees.

Two minor writing points that the authors will want to fix: it should be "may host stable" rather than "may hosts stable" at the top of page 6, and there is a typo in the sentence added to the abstract (should be "indicates" rather than "indicate").

Incidentally, I think that the added sentence may be read with a different meaning than the authors intend. It seems to imply that the result contradicts some (claimed) theorem. I assume what they mean to say is more like, "This result indicates that clean-limit optical responses are common in superconductors beyond the straightforward case of broken inversion symmetry." Anyway, I do not think this question of wording should delay the paper's publication.

Reviewer #2 (Remarks to the Author):

I studied the authors' reply to all questions raised by myself as well as by the other referees. I find their response and the adjustments made to the manuscript appropriate, and recommend the publication of this work in the present form.

Reviewer #3 (Remarks to the Author):

The authors have carefully addressed the reviewers' comments. The revised manuscript is now clear also for non-experts. Since this work represents significant progress in our understanding of optical properties of superconductors and is of direct relevance for experiments on FeSe, I recommend its publication in the present form.

Response to Reviewers

We appreciate all reviewers for recommending our paper for publication. We thank reviewer #1 for correcting our typos and giving a helpful comment again. We have corrected the typos and revised our abstract as suggested.